# Epidemiology and Incidence of Pediatric Concussions in General Aspects of Life

**DOI:** 10.3390/brainsci9100257

**Published:** 2019-09-27

**Authors:** Chang Yaramothu, Arlene M. Goodman, Tara L. Alvarez

**Affiliations:** 1Department of Biomedical Engineering, New Jersey Institute of Technology, Newark, NJ 07102, USA; chang.yaramothu@njit.edu; 2Saint Peter’s Sports Medicine Institute, Saint Peter’s University Hospital, Somerset, NJ 08873, USA; agoodman@saintpetersuh.com

**Keywords:** concussions, physical education, concussion epidemiology, adolescent concussions

## Abstract

Background: Concussions are one of the most common head injuries acquired within the pediatric population. While sport-related concussions are well documented, concussions within other aspects of a child’s life are not as well researched. The purpose of this study is to examine the incidence of a large pediatric concussion population in a broad range of daily activities. Methods: Patients’ gender and nature of injury were extracted from 1408 medical records of patients who were diagnosed with a concussion at Saint Peter’s Sports Medicine Institute. Statistical analyses were conducted for activities and environmental settings using chi-squared tests. Results: Concussions were most prevalent in organized sports (53.3%), followed by injuries within the following settings: school (16.5%), recreational (6.7%), motor vehicle collisions (6.6%), home (5.5%), and other (11.3%). Specifically, soccer (12.9%), school physical education (PE) class (10.6%), and football (9.8%) subcategories recorded the most incidences of concussion. For the PE class cohort (*n* = 149), significantly more females were diagnosed with a concussion compared to males (*p* < 0.001). Conclusions: PE-related concussions had the second highest incidence rate after organized sports. A significant gender difference was observed in PE class. Awareness about concussions and methods to reduce the risk of concussion is suggested for PE classes.

## 1. Introduction

Concussion is a common traumatic head injury that is most often associated with contact sports or battlefield injuries [1,2]. Previous studies have focused on incidence of concussions in specialized sports, such as ice hockey [3] or football [4,5], and specific pediatric/youth settings, such as organized high school [6] or collegiate sports [1,7,8]. Studies have generally concentrated on the percentage of athletes that incur concussions playing a specific sport in a general geographical location, as well as which mechanisms of that sport are the leading cause of concussions. Studies have also shown that the prevalence of concussion diagnosis increased by 71% in the 10- to 19-year-old age group between 2010 and 2015 [9]. The increase in concussion diagnosis is probably due to improved awareness of concussion injuries by the general population, and by caregivers and athletic trainers/coaches becoming more vigilant in seeking medical care when a child presents with symptoms associated with concussion post-injury. Studies of the past, however, have typically not examined the incidence of concussions acquired within the general pediatric population in a variety of settings other than sports.

The diagnosis of concussion is on the rise, and the exact epidemiology is still not completely understood. Many studies report that females have a greater incidence of sports-related concussion compared to males in organized sports [10,11,12,13]. This purpose of this study is to examine the mechanism of pediatric concussion as a function of gender during a four-year span for all participants who sought treatment for concussion at Saint Peter’s Sports Medicine Institute in Somerset, New Jersey.

## 2. Materials and Methods

### 2.1. Study Design

The study design was retrospective, and methods will be discussed in terms of concussion population sample, classification of concussions, rationale for concussion incidence groupings, and statistical analyses. The Saint Peter’s University Hospital Ethics Committee for the Protection of Human Subjects in Research approved this retrospective study, in accordance with the United States Code of Federal Regulations 21CFR 56.110.

### 2.2. Setting and Participants

The data were extracted from records of pediatric concussion participants by one of the authors (A.G.), a pediatric sports medicine physician, who diagnosed all the participants with a concussion. The concussion setting (nature of activity when a concussion was acquired) and the gender of the patient were collected. The data presented in this study represents all the participants that sought treatment at the Saint Peter’s University Hospital (SPUH) Sports Medicine Institute (SMI) in Somerset, New Jersey, between 15 September 2014 and 31 May 2018 (~4 years). SMI is located in Central New Jersey and is easily accessible to at least a third of the state’s population of 2 million pediatric-age individuals [14,15]. Only the participants who were diagnosed with a concussion were included in this study, all other participants were excluded. This study analyzed existing data that did not contain any personal identifiers, and was approved by the Saint Peter’s University Hospital Institutional Review Board for an exemption. A total of 1408 participants were examined, of which 56% (*n* = 790) were female.

### 2.3. Variables and Data Measurement

The primary variables were mechanism of injury and gender. Incidences of concussion were organized into five primary categories: organized sports, school setting, home setting, motor vehicle collisions, and recreational activities. Concussions that were not categorized into the prior five categories were grouped as “other”. All mechanism of injury and gender data were classified within a single category. No data were included in two or more categories. The categories of organized sports and school setting were subdivided into subcategories for an analysis of specific concussion incidences. The following subcategories are included under organized sports: soccer; football; basketball; lacrosse; cheerleading; combat sports, which includes mixed martial arts (MMA); boxing and wrestling; ice hockey; field hockey; softball; and other organized sports. The sports squash, tennis, golf, ultimate frisbee, etc., were grouped into the “other organized sports” subcategory of organized sports. The concussions resulting from a school setting were analyzed with two methods. One analysis for the school setting cohort examined incidence of concussion, investigating gender as a function of the child’s type of school—specifically, primary, middle, or high schools. The other analysis for the school-setting cohort examined incidences of concussions in physical education (PE) classes as a function of gender. This sub-category includes the study of all activities within PE class, such as PE basketball and PE volleyball. No data were classified in more than one sub-category or cohort.

The motor vehicle collisions (MVC) cohort includes concussions acquired while inside a motor vehicle (either as a driver or passenger), or an impact by a motor vehicle as a pedestrian. The home-setting cohort includes all injuries and collisions which lead to a concussion diagnosis within a patient’s home. These injuries range from falling in the shower, slipping on ice near the home, or a blunt injury to the head while colliding with a door, to what many patient’s parents described as “horsing around”. The recreational cohort includes head injuries while bicycling, skateboarding/hoverboarding, snowboarding, sledding, tubing, ice skating, trampolining, ziplining, horseback riding, alcohol related injuries, and injuries acquired in amusement parks. Finally, the following activities and settings are examples that could not be grouped into any of the five primary categories, and were listed within the “other” cohort: tripping while casually walking, slipping on ice (not near the home such as a shopping area), fainting, head injury while entering a car, being involved in a physical altercation, and having objects (i.e., chairs) thrown at subject.

### 2.4. Concussion Diagnostic Criteria

The criteria for the diagnosis of a concussion included the following: (1) patient sustained a direct impact to the head, face, neck, or elsewhere on the body that could result in an impulsive force transmitted to the head; and (2) the presentation of at least one symptom (headache, dizziness, difficulty concentrating, etc.) within 24 h of sustaining the injury during cognitive or physical exertion. Participants who had sustained a head injury but were (1) symptom-free the following day, (2) could read for longer than 30 min, (3) could do homework for two or more hours, and (4) could perform moderate to heavy physical exertion without symptoms were not diagnosed with a concussion. These subjects were excluded from this analysis.

#### Power Size and Statistical Analyses

Statistical power for an effect size between 0.1 and 0.2, with 80% power and alpha = 0.05, yields a sample size between 392 and 1568 participants [16]. With 1408 participants, even small effect size differences will be observed within this studied population. The chi-squared test assumed that the expected observation was utilized to determine whether a significant gender difference was observed within a cohort. For example, since the population studied here had 56% females within the entire population, then it would be expected that if no gender difference occurred, about 56% of each cohort would be female. The null hypothesis states that all categories have the same proportions. Statistical significance was defined to be a significant level (*p*-value) of less than 0.05 for the chi-squared test. The significance *p*-value was corrected with Bonferroni’s correction for multiplicity of tests, in order to avoid Type I errors. Since a total of 10 statistical comparisons were performed, the significance *p*-value was lowered to 0.005 to correct for multiple comparisons. Gender differences were computed in the following cohorts, due to the expectation that male and female participation was about equivalent: all organized sports, PE for schooling levels (primary, middle, and high), organized soccer, and organized basketball, as well as the high school setting, middle school setting, primary school setting, home setting, recreational activities, and motor vehicle collisions. There were no missing data to account for within this study. Statistical analyses were computed utilizing IBM SPSS Statistics 20 package.

## 3. Results

The following findings will be presented: incidences of all recorded concussions from the primary and sub-categories, concussion percentages in each category, top-five categories of activities where concussions were observed, and an incidence rate of concussion as a function of gender comparison.

The numbers and percentages of concussions recorded in each cohort are reported in Table 1 and Figure 1. The organized sports cohort recorded the highest number of concussions with 53.3% (*n* = 751). The remaining concussions were attributed to concussions within the school setting (*n* = 232, 16.5%), recreational activities (*n* = 95, 6.7%), motor vehicle collisions (*n* = 93, 6.6%), and the home setting (*n* = 78, 5.5%). A few (*n* = 159, 11.3%) of the concussions did not fall into any of the five primary categories, and are listed as “other”.

Soccer was observed to have the highest number of concussions, at 24.2% (*n* = 182), within the organized sports cohort (*n* = 751). Football and basketball followed at 18.4% (*n* = 138) and 11.1% (*n* = 83), respectively. These three sports accounted for approximately 54% of all organized sports-related concussions. Lacrosse (*n* = 61, 8.1%), cheerleading (*n* = 61, 8.1%), and combat sports (*n* = 47, 6.3%) had the next three highest incidences of sports-related concussions. The combat sports of wrestling, boxing, and MMA were grouped together because of the similar nature of the activities. The final three sports to have the highest incidence of concussions were ice hockey (*n* = 33), softball (*n* = 29), and field hockey (*n* = 26), at approximately 2% each. Other organized sports-related concussions accounted for the remaining 6.5% (*n* = 91) of the concussions in this category.

For the school setting primary category (*n* = 232), there were two different types of subcategorizations of the data performed (see Table 1). The first divided the concussions acquired within a school setting by type of school—specifically, high, middle, and primary school. The second investigated only the school-setting concussions that occurred within PE classes for high, middle, and primary schools combined. For the first analysis, high schools were recorded to have the highest incidence of concussions, accounting for 43.1% (*n* = 100) of all concussions in a school setting. Middle (31.9%, *n* = 74) and primary schools (25.0%, *n* = 58) had comparatively lower incidence of concussions. For the second analysis, concussions from PE class accounted for 64.2% (*n* = 149) of all concussions that were recorded in a school setting. PE basketball (15.1%, *n* = 35) and PE volleyball (11.6%, *n* = 27) accounted for the highest number of incidents of concussions in a school setting; this translates to 23% and 18% of all recorded PE incidences of concussions, respectively.

Gender differences among the organized sports category are reported within Table 2. Organized sports that traditionally have higher male participation, such as football, combat sports, ice hockey, or lacrosse, had higher incidence of male concussions at 99% (*n* = 136), 89% (*n* = 41), 82% (*n* = 27), and 64% (*n* = 39), respectively, when compared to their female counterparts. Conversely, sports with traditionally higher female participation in the state of New Jersey, United States such as cheerleading, field hockey, and softball had a female concussion incidence rate of 100%. In the remaining sports of soccer and basketball, where male and female participation is assumed to be similar, females had higher incidence of concussions at 57% (*n* = 103) and 59% (*n* = 49), respectively. These higher observed female incidences of concussion in soccer (χ^2^ = 0.022, *p* = 0.88) and basketball (χ^2^ = 0.439, *p* = 0.51) are, however, not statistically different from the gender proportions of the entire study population. The total number of males who experienced a sport-related concussion in the more popular organized sports (*n* = 356) was compared to the number of females who experiences a sport-related concussion in the more popular organized sports (*n* = 304). When these rates were compared to the gender proportions of the entire study population (expected outcome), the higher incidence of male concussions was statistically different (χ^2^ = 16.9, *p* < 0.001).

Figure 2 reported a comparison of the overall top five mechanisms of injury for concussions. The highest number of concussions were recorded in soccer (12.9%, *n* = 182) as an organized sport. The second highest incidence of concussions were documented in a school setting, specifically, during PE class, accounting for 10.6% (*n* = 149) of all the concussions. Any physical activity that was part of the PE curriculum, which included an array of diverse physical activities, including soccer, basketball, volleyball, etc., was pooled into the “school PE” cohort. The only other organized sport in the top five was football (9.8%, *n* = 138), in third place. The fourth and fifth highest incidence of concussion were recorded by participants who incurred a concussion while performing recreational activities (6.7%, *n* = 95) and those involved in motor vehicle collisions (6.6%, *n* = 93).

Figure 3 reported activities in which male and female participation would normally be assumed to be similar, if not equivalent. The three primary categories of recreational activities (65%, *n* = 40), home setting (57%, *n* = 40), and motor vehicle collisions (71%, *n* = 62), all had higher female incidences of concussion compared to males. However, the higher rate of female concussions was only statistically significant in motor vehicle collisions (χ^2^ = 8.55, *p* = 0.003), and not the other two categories.

The trend of higher female concussion incidence follows in the school setting. Females accounted for 66% (*n* = 152) of all concussions that incurred in a school setting. When examining the individual school levels, females only have a statistically higher proportion of concussions (85%, *n* = 85) in high school (χ^2^ = 34.1, *p* < 0.001). Middle schools had a higher rate of female concussions (59%, *n* = 44), while primary schools showed a higher incidence of male concussions (60%, *n* = 35); however, neither difference showed statistical significance (*p* > 0.05 and *p* > 0.01, respectively). Females, additionally, have a higher proportion of concussions (χ^2^ = 18.6, *p* < 0.001) when investigating the cohort of concussions reported during PE class (73%, *n* = 108) when all three school levels were examined. The two PE sports that had the highest incidence of concussions, PE volleyball and PE basketball, also had higher female incidence of concussion at 85% (*n* = 23) and 77% (*n* = 27), respectively.

## 4. Discussion

### 4.1. Organized Sports

The results of this retrospective study are mostly consistent with previous studies in relation to gender differences in the incidence of concussions [6,10,17,18]. Overall, for this study more females sought treatment for a concussion compared to males. Due to the limited regional scope of this retrospective study, the simple conclusion that females in general are more prone to concussion compared to males is not definitive. One possible explanation could be a female preference to seek care from a female practitioner rather than a male practitioner. This explanation, however, cannot be supported through any recent studies, and a further in-depth analysis of the additional data points reveals other trends. When the entire organized sports cohort of this study is examined, the recorded number of male concussions (52%) is statistically different (*p* < 0.001) than the number of female concussions when compared to the total gender proportion of the study. This difference, however, could be a false positive, as there simply could have been more male athletes compared to females. This false positive could also be attributed to the larger number of sports that have predominately male participants, such as football, combat sports, and ice hockey.

The analysis of organized sports that are presumed to have equal gender participation would reduce the false positive rate. As anticipated, sports that are considered to have predominately male participants, such as football, lacrosse, combat sports (including MMA, wrestling, and boxing), and ice hockey, had higher incidences of male concussions. Conversely, sports with traditionally predominant female participation in New Jersey, such as cheerleading, field hockey, and softball, had higher incidences of female concussions. However, when sports with equal male and female participation, such as soccer (57%) and basketball (59%), were examined, females were observed to have a higher incidence of concussions; however, these higher female observations were not statistically different (*p >* 0.5) from gender proportions of the total study population. It is possible that females are getting concussed more frequently than males; however, due to the scope of this study, only comparison of the observed incidences with the study population’s proportions can be conducted. Many prior studies have shown that females have higher proportions of concussions in soccer and basketball [10,17,19].

Previous studies have consistently observed the highest instances of concussions in full-contact sports, such as football or lacrosse [6,17,20]. This study, however, identified a different order of highest incidences of concussions among the pediatric population: soccer, football, basketball, lacrosse, cheerleading, combat sports, ice hockey, softball, and field hockey. There are a few possible rationales for a higher incidence of observed concussions in soccer compared to football. First, soccer is a year-round sport, compared to football, which is predominately a fall sport; this results in soccer having a higher number of opportunities for concussions to occur. Additionally, recent general media reports from New Jersey [21,22,23] and across the United States [24,25,26] describe a decline in organized football participation, speculated to be linked to public concerns over safety.

### 4.2. Concussions in Schools

Sports-related concussions in organized sports was the largest cause of concussions, as observed in this study, accounting for 53% of all recorded concussions. However, the second leading cause was previously unclear, and is shown in this study to occur in a school setting for pediatric and adolescent age groups. Injuries in a school setting were those that occurred either during PE class, in a classroom, during recess, or in the hallway. Consistent with the general trend of the highest proportion of concussions being sports-related, most of the concussions in the school were also sports-related; specifically, during PE (64% of the school setting cohort). The incidences of concussion during school PE accounted for 11% of the total study’s concussions—whereas organized soccer accounted for 13% of the total concussions, and organized football accounted for only 10% of the total concussions, which is fewer than school PE. Additionally, unlike in organized sports, females in PE were diagnosed with a concussion significantly more often than males (*p* < 0.001).

One rationale for the high rates of incidence of concussion during school PE compared to organized sports could be the combination of differences, including varying growth rates of adolescents, their maturity, and gender. The larger disparity of physical sizes within a school setting may lead to an increased opportunity for an injury. Additionally, variations in size and differences in gender could also explain the high number of female concussions in PE classes, especially in PE volleyball and PE basketball. This rationale could also potentially explain why the gender difference in concussions is statistically different and greater during PE sports compared to organized sports, which are generally not co-educational.

Another reason could be the extent and type of training/coaching that is administered to the students in PE class, compared to the pediatric/adolescent athletes. Coaches in organized sports have adequate time to teach the young athletes proper techniques and measures for injury prevention. However, in school PE the primary objective for the educators is to aid the students in being physically active in the limited time provided in a school day or week. This preference for any form of physical activity makes it difficult for every student to be taught proper techniques and measures for injury prevention, especially when there is a lower interest in participation for students in PE compared to the pediatric/adolescent athletes as they play their sport. It also makes it difficult for the educators to provide in-depth guidance when the teacher-to-student ratios can be in excess of 1:30. Finally, physical education in the state of New Jersey is mandatory; this results in multiple classes being grouped together in a single gymnasium at times of inclement weather, raising the chances for an injury to occur. This does not mean physical education should be decreased; prior studies unanimously agree on the importance on physical education for the pediatric population [27,28,29]. Rather, it is beneficial to place an importance on the perils of injuries, including concussions, not only in organized sports but in any environment where sports are being played, including PE class. By extension, it would be prudent to take the time to teach everyone proper techniques and injury prevention.

Only one other previous study, conducted by Nelson et al. [30] between 1997 and 2007, has examined any results that are similar to the ones observed in this retrospective investigation. Nelson et al. examined any injuries (including head injuries, lacerations, fractures, soft-tissue injury, etc.) treated in emergency departments across the United States that were incurred during physical education class. This previous study identified the greatest number of all injuries in middle school, primary school, and high school, in descending order. Our study, in contrast, identified high schools as having the highest number of concussions, followed by middle and primary schools. Our study also noted statistically higher incidences of female concussions at the high school levels (*p*< 0.001), but not at the middle (*p* = 0.58) or primary (*p* = 0.02) school levels. A potential explanation for this difference could be the rate at which males and females differentiate in their growth in high school between the ages of 13 and 18.

### 4.3. Concusions Outside of School or Organized Sports

After organized sports-related concussions and school-related concussions, the highest number of concussions were reported in participants who were participating in recreational activities (7%), involved in motor vehicle collisions (7%), and finally while participants were home (6%). Although injuries in the home setting and during recreational activities showed no statistical differences in gender, MVCs (*p* < 0.001) showed a statistically higher proportion of female concussions. Most prior studies on the epidemiology of concussions do not examine these other aspects of life where concussions can occur. These concussions may not always be preventable, as they are occurring during everyday activities. It is important to record them and make the information available to everyone. This will hopefully change the perception of concussions being only from a battlefield or athletic trauma, to resulting from everyday injury as well. Concussions can be incurred in a school setting or during everyday activities for the pediatric population, such as taking a skateboard down the street or while at an amusement park.

## 5. Conclusions

Studies from other regions throughout the world are needed to determine whether the trends observed here generalize to other populations. The trends and gender differences observed in this study, such as the higher prevalence of female concussions during school PE and motor vehicle collisions, have the potential to be observed in other regions as well. This knowledge could potentially raise awareness and make the general population more vigilant, hopefully leading to fewer concussions in the future and encouraging more people to seek treatment after a suspected concussion. It is important to keep track of the incidences of concussions during organized sports, but it should be equally as important to track concussions in all other aspects of life. Although organized soccer and basketball did not yield gender differences that were statistically significant, a longitudinal study that does not rely on expected outcomes could yield results consistent with previous studies showing higher rates of female concussions.

## Figures and Tables

**Figure 1 brainsci-09-00257-f001:**
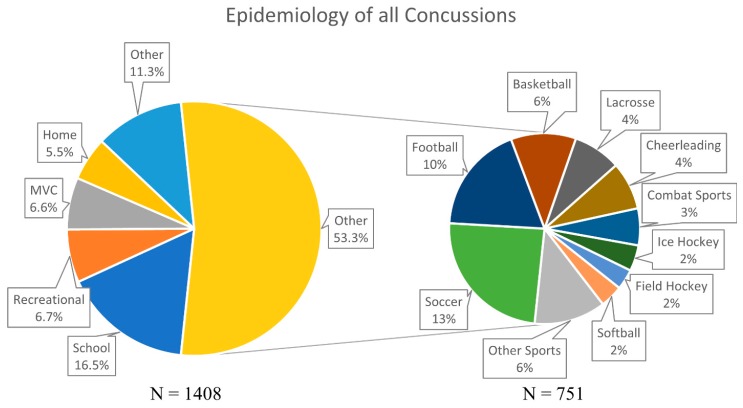
Detailed percentages of all concussions, divided into organization sports and non-organized sports. MVC: Motor Vehicle Collisions.

**Figure 2 brainsci-09-00257-f002:**
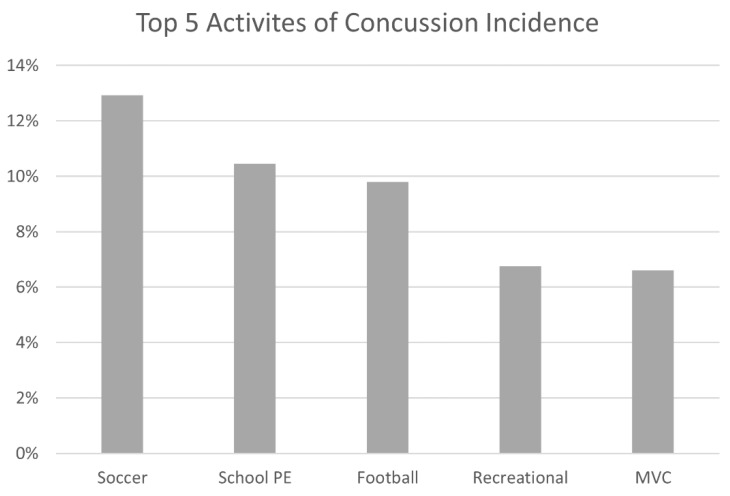
Top five categories of concussions. These categories include soccer as an organized sport, school physical education (PE), football, recreational, and motor vehicle collisions (MVC).

**Figure 3 brainsci-09-00257-f003:**
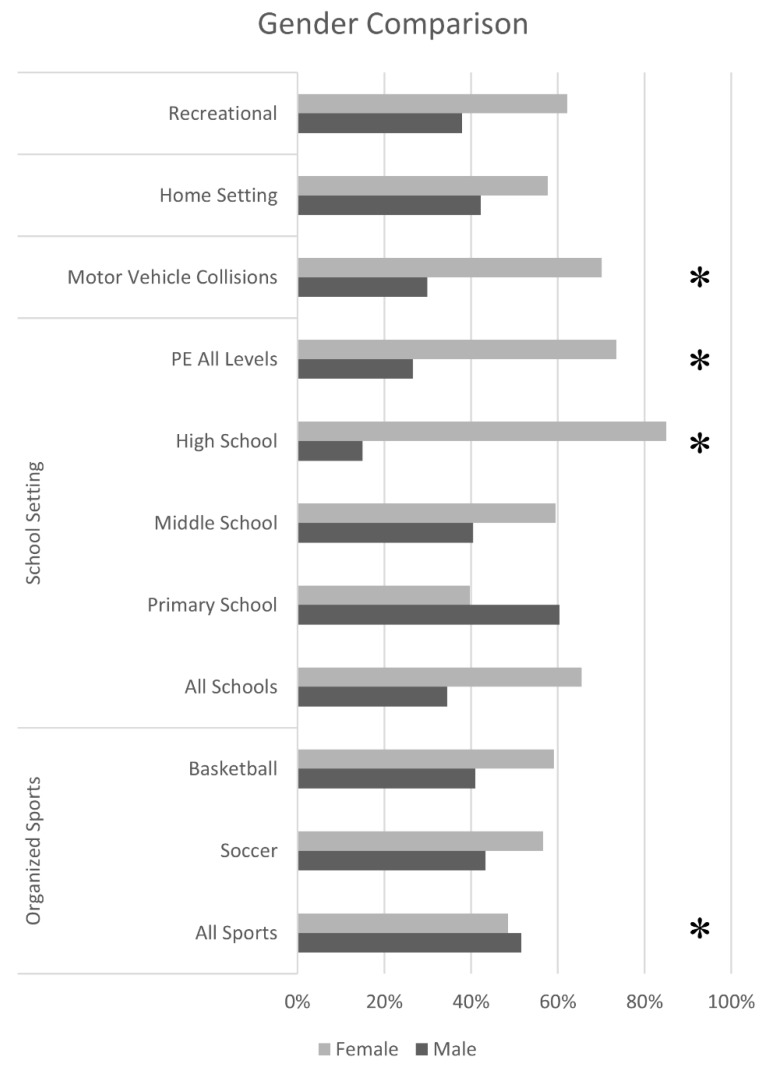
Gender comparison. Only the categories where male and female participation would be expected to be similar are listed. The asterisks denote a significant chi-square test by gender (*p* < 0.005).

**Table 1 brainsci-09-00257-t001:** Epidemiology of all concussions. Wrestling, boxing, and mixed martial arts (MMA) were grouped into the larger combat sports category. Cheerleading and color guard were grouped together into cheerleading. The five primary categories (organized sports, school setting, recreational activities, motor vehicle collision, and home setting) of mechanism of injury for concussions are shown in dark grey cells. The light grey cell are the sub-categories of one of the primary categories.

Epidemiology		No. of Concussions	% of Total Concussions	% of Each Primary Concussion Categories
Total Concussions		1408	100	
Organized Sports		751	53.3	100
	Soccer	182	12.9	24.2
	Football	138	9.8	18.4
	Basketball	83	5.9	11.1
	Lacrosse	61	4.3	8.1
	Cheerleading	61	4.3	8.1
	Combat sports	47	3.3	6.3
	Ice Hockey	33	2.3	4.4
	Softball	29	2.1	3.9
	Field Hockey	26	1.8	3.5
	Other Organized Sports	91	6.5	12.1
School-Setting		232	16.5	100
Schooling Levels	High School	100	7.1	43.1
	Middle School	74	5.3	31.9
	Primary School	58	4.1	25.0
In Physical Education Class	For High, Middle and Primary Schooling Levels	149	10.6	64.2
	Basketball	35	2.5	15.1
	Volleyball	27	1.9	11.6
	Other School Setting Activities	87	6.2	37.5
Recreational Activities		95	6.7	
Motor Vehicle Collision		93	6.6	
Home-Setting		78	5.5	
Other		159	11.3	

**Table 2 brainsci-09-00257-t002:** Gender differences in organized sports.

Organized Sport	No. of Males (%)	No. of Females (%)
Soccer	79 (43%)	103 (57%)
Football	136 (99%)	2 (1%)
Basketball	34 (41%)	49 (59%)
Lacrosse	39 (64%)	22 (36%)
Cheerleading	0 (0%)	61 (100%)
Combat Sports	41 (89%)	6 (11%)
Ice Hockey	27 (82%)	6 (18%)
Field Hockey	0 (0%)	29 (100%)
Softball	0 (0%)	26 (100%)
Total	356	304

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
