# Peer review of "Epidemiology and Incidence of Pediatric Concussions in General Aspects of Life"

_brainsci, 2019, doi:10.3390/brainsci9100257_

Round 1
Reviewer 1 Report
The authors present a well articulated and clear study examining epidemiological rates of concussion in a pediatric population. The study is well presented, the statistical methods appear appropriate and the classifications in general make sense for interpretation of the data. Overall the data are supportive of previous findings in terms of epidemiology of concussion. This study is novel in that it addresses the distribution in areas of life that are perhaps not frequently studied. A few minor corrections and clarifications would be helpful for this study.
Overall, the data is important to highlight the risk of concussion across the 3 levels of school (primary, middle and highschool). It is perhaps not surprising that higher rates of concussion are observed in highschool due to larger body sizes which translate to greater inertial mass, acceleration and impact force. The authors conclude by stating that the data would be helpful to raise awareness about the potential for concussion in everyday activity and not only in organized sports or military settings. It is always good to raise awareness about concussion. However, the rate of concussion among these population cohorts is more relevant towards supporting these statements but perhaps not available with the given data set.
Items for clarifications:
Slipping on ice close to home was included in the home category but also in the "other" as slipping on ice. Is there potentially a double count in data sets?
Soccer is listed as one top cause of concussion in organized sports. Was there any overlap in sports played in school? ie. Is soccer part of the PE curriculum?
Table 2. "Filed hockey" typographical error
Figure 2. Figure legend erroneously indicates 'basketball' instead of 'recreational'
Author Response
Reviewer #1:
Thank you for your review. Below is a point by point response to your comments.
Comments and Suggestions for Authors
The authors present a well-articulated and clear study examining epidemiological rates of concussion in a pediatric population. The study is well presented, the statistical methods appear appropriate and the classifications, in general, make sense for interpretation of the data. Overall the data are supportive of previous findings in terms of the epidemiology of concussion. This study is novel in that it addresses the distribution in areas of life that are perhaps not frequently studied. A few minor corrections and clarifications would be helpful for this study.
Overall, the data is important to highlight the risk of concussion across the 3 levels of school (primary, middle and high school). It is perhaps not surprising that higher rates of concussion are observed in high school due to larger body sizes which translate to greater inertial mass, acceleration and impact force. The authors conclude by stating that the data would be helpful to raise awareness about the potential for concussion in everyday activity and not only in organized sports or military settings. It is always good to raise awareness about concussion. However, the rate of concussion among these population cohorts is more relevant towards supporting these statements but perhaps not available with the given data set.
Items for clarifications:
COMMENT 1 = Slipping on the ice close to home was included in the home category but also in the "other" as slipping on ice. Is there potentially a double count in data sets?
RESPONSE 1 = We appreciate the comment. The data were not counted twice. There were 8 patients who slipped on ice: 4 near the home and 4 away from home which was categorized as ‘other’. When slipping on ice was close to the home, it was categorized as being within the home setting. However, when slipping on ice was not in the home setting such as near a shopping area then it was categorized as ‘other’. To improve the clarity of the manuscript, the following sentences have been added. “All mechanism of injury and gender data were classified within a single category. No data were included in two or more categories.” Lines 73-74 page 2. “No data were classified in more than one sub-category or cohort.” Lines 85-86 page 2. In addition, the following sentence was modified. “Finally, the following activities and settings are examples that could not be grouped into any of the five primary categories and were listed within the ‘other’ cohort: tripping while casually walking, slipping on ice (not near the home such as a shopping area), fainting, head injury while entering a car, being involved in a physical altercation, and having objects (i.e. chairs) thrown at subject.” Lines 94-98 page3
COMMENT 2 = Soccer is listed as one top cause of concussion in organized sports. Was there any overlap in sports played in school? ie. Is soccer part of the PE curriculum?
RESPONSE 2 = There were two types of soccer-related concussions. Soccer as an organized sport (not within the PE curriculum) was the top cause of concussion. Within the PE curriculum, soccer was one of the many potential physical activities that students participated within. All physical activities regardless of the type that were part of the PE curriculum were grouped together including soccer. The manuscript now states: “Any physical activity that was part of the PE curriculum which included an array of diverse physical activities including soccer, basketball, volleyball, etc. was pooled into the School PE cohort.” Line 187-189 page 6.
COMMENT 3 = Table 2. "Filed hockey" typographical error
RESPONSE 3 = Done. Line 182 page 6
COMMENT 4 = Figure 2. Figure legend erroneously indicates 'basketball' instead of 'recreational'
RESPONSE 4 = Done. Line 196 page 6
Reviewer 2 Report
In this manuscript, the authors conducted a retrospective study on the incidence and epidemiology of pediatric concussions in daily activities, such as organized sports, PE class at school, recreational activities, home-setting activities, etc. After examining 1408 medical records of patients with concussion, they showed that PE related concussions had the second largest incidence rate right after organized sports. Additionally, they found that the incidence of concussions at PE class was significantly higher in females compared to males.
Overall, this study was well designed, and the results are convincing and discussed well.
Author Response
Reviewer #2:
Thank you for your review. Below is a point by point response to your comment.
Comments and Suggestions for Authors
COMMENT 1 = In this manuscript, the authors conducted a retrospective study on the incidence and epidemiology of pediatric concussions in daily activities, such as organized sports, PE class at school, recreational activities, home-setting activities, etc. After examining 1408 medical records of patients with concussion, they showed that PE related concussions had the second-largest incidence rate right after organized sports. Additionally, they found that the incidence of concussions in PE class was significantly higher in females compared to males.
Overall, this study was well designed, and the results are convincing and discussed well.
Response 2 = Thank you. We appreciate the review.